# Road Transport and Its Impact on Air Pollution during the COVID-19 Pandemic

Katerina Vichova *[iD], Petr Veselik [iD], Romana Heinzova [iD] and Radek Dvoracek [iD]

Faculty of Logistics and Crisis Management, Tomas Bata University in Zlín, Studentské náměstí 1532, 686 01 Uherské Hradiště, Czech Republic; veselik@utb.cz (P.V.); rheinzova@utb.cz (R.H.); r1_dvoracek@utb.cz (R.D.)
* Correspondence: kvichova@utb.cz; Tel.: +420-576032088

**Abstract:** This paper discusses the impact of the COVID-19 pandemic on air pollution. Many urban inhabitants were confined to their homes during the lockdown. This had an impact air pollution, due to a reduction the number of vehicles being operated in cities. People also limited the number of visits to shopping centers; additionally, sports venues were closed and cultural events cancelled. The COVID-19 pandemic therefore had a positive impact on air pollution. Several studies from around the world confirm this. The research presented here is based on hourly measurements of $PM_{10}$ and $NO_2$ concentrations measured in background ambient air at a specific intersection located in Uherske Hradiste, Czech Republic. The aim of the paper is to confirm or exclude the hypothesis that the measured concentrations of $PM_{10}$ and $NO_2$ pollutants were lower during 2020 than in 2019, when states of emergency related to the COVID-19 pandemic were declared. The data were aggregated into monthly subsets and statistically analyzed. The data was graphically visualized and evaluated by means of exploratory data analysis. To compare the pollution levels in individual months, a parametric statistical analysis (two-sample *t*-test) was used. A statistically significant reduction was observed in the measured concentrations in 2020 compared to 2019 during periods when states of emergency were declared.

**Keywords:** air pollution; COVID-19 pandemic; sustainable transport; particulate matter $PM_{10}$; nitrogen dioxide; two-sample *t*-test

## 1. Introduction

Transport has become an essential part of our day. Its purpose is to move people and things. For the proper functioning of the state, it is necessary to ensure the operation of critical infrastructure sectors. Transport can be included among these sectors [1]. The European Council specified four subsectors: road, rail, air, and waterway transport [2]. Rehak et al. state that road transport currently has an irreplaceable role in the European transport network [3]. The field of transport brings not only advantages but also associated risks associated. One of the most significant risks in terms of transport can be environmental risks. One of the environmental risks in the field of transport is air pollution by exhaust gases which endanger human health and cause global warming. Hoterova and Dvorak highlight that this issue, which includes the production of greenhouse gases, has resonated significantly in recent years [4]. They add that road transport is one of the largest producers of emissions. Each country in the world should strive to reduce emissions. The European Commission, in 2021, adopted a set of proposals to make the EU's policy-making climate fit for reducing net greenhouse gas emissions by at least 55% by 2030 compared to 1990 levels, focusing not only in the field of transport [5]. However, the whole world was affected by the COVID-19 pandemic, which affected all sectors. As many as one-third of employees worked from a home office during the lockdown.

As a result, the number of vehicles on the roads decreased, but so did the use of urban public transport. In response to the current situation, the research team's hypothesis

centered on whether the impacts on transport sector affected levels of air pollution during the COVID-19 pandemic. Several studies confirmed that there was a reduction of emissions during the pandemic [6–12]. However, there are not many studies focused on the Czech Republic. Among the limits of this research is the fact that we only verified the hypothesis based on a selected intersection in Uherske Hradiste, Czech Republic. The findings might not, therefore, apply to the whole of the Czech Republic. This research was undertaken as part of the Logistics Risk Management project, which aims to highlight the environmental risk of road transport. The aim of the paper is to confirm or exclude the hypothesis that measured concentrations of nitrogen dioxide ($NO_2$) and particulate matter with an aerodynamic diameter less than or equal to 10 micrometers ($PM_{10}$) pollutants would be lower during individual months when states of emergency related to the COVID-19 pandemic were declared.

## 2. Theoretical Background

Critical infrastructure networks such as transport are essential for the functioning of a society and the economy. When functioning regularly, transport networks are depended upon by on millions of commuters and travelers worldwide every day [13]. Due to its geographical location, the Czech Republic has a unique position within the Central European region. The globalization of world trade has caused enormous demand for transport and logistics services, which is also the case in the Czech Republic [14]. The logistics system has been increasingly recognized as one of the fundamental driving forces for economic growth [15]. A developed logistics infrastructure facilitates backward and forward linkages in global trade and helps to manage the business cost of accessing markets [16]. One of the critical elements of the logistics system is transport. Transport provides access to employment opportunities, raw materials, goods, and services and enables the movement of populations. Vavrek and Becica define transport as the deliberate and organized relocation of objects and persons by means of various types of transport on railways and roads [17]. According to Patrman et al., land transport provides a necessary service for the functioning of society. The importance of these elements depends on the level of performance they provide, i.e., their traffic-carrying capacity and traffic intensity [18]. Transport ranks among the fastest developing sectors of the national economy in developed countries [19].

A consequence of such fast development has been the adverse environmental effects, which many governmental officials (within countries, regions, and municipalities) have tried to resolve by implementing new policies aimed at restricting the massive development of passenger motoring [20]. Transport greatly influences the sustainability and quality of life in cities [21]. Rapid urbanization and increasing numbers of trips transporting both people and cargo have worsened cities' air quality, presenting a significant challenge for urban environmental management [22]. The global balance of transport activities has only recently been systematically analyzed, after its constant growth created a context of more significant concern about the deterioration of the environment and in the wake of increasing awareness of the scarcity of a considerable number of resources [23]. The modern transport system includes a large number of personal vehicles. Given the high numbers, traffic congestion, fuel consumption, and greenhouse gas emissions have become a serious problem [24]. Babak defines the synergistic consequences of transport activities and considers their various direct and indirect impacts on ecosystems, which are frequently unpredicted. Climate change, which has both complex causes and consequences, is the cumulative impact of several natural and anthropogenic factors, in which transport has a role [25].

### 2.1. Transport and the Environment

Air pollution is the most significant environmental risk to human health and the second biggest environmental concern for Europeans after climate change [26]. The transport sector represents almost a quarter of greenhouse gas emissions, and it is considered the

main cause of air pollution in cities [27]. Recent reports by the European Environmental Agency (EEA) point out that the transport sector accounts for about 47% of nitrogen oxide and 13% of particulate matter emissions across 33 EEA countries [28]. For the second consecutive year, the average $CO_2$ emissions from new passenger cars increased in 2018 and reached 120.8 $gCO_2$/km. The main factors contributing to that increase include the growing share of petrol cars with new registrations, particularly in the sport utility vehicle (SUV) segment [29]. Several substances harmful to health and the environment are released into the atmosphere by motor vehicles, mainly due to the burning of fossil fuels, causing air pollution [30]. Road transport and the resulting vehicular emissions have become one of European nations' leading challenges [31]. Road transport accounts for over 20% of the carbon dioxide emissions in Europe and is the only source of greenhouse gas emissions that has steadily trended upwards since 1990 [28]. In Europe, about one-fifth to one-third of the urban population had been exposed to $PM_{10}$ concentrations above EU and WHO reference levels by 2011 [32]. The impacts of PM emissions on climate and human health have become a subject of significant concern for the scientific community and public powers worldwide in recent years [33]. Generally, the sources of $PM_{10}$ are heating, fieldwork, agriculture, iron production, and transport, among others. In urban areas, motor vehicles represent the primary source of magnetic PM. It is mainly emitted as a result of the abrasion of disk brakes and from fuel combustion residues emitted by diesel and gasoline exhausts [34]. Based on the location of our testing station, we expected that the primary source of $PM_{10}$ would be transport. The primary anthropogenic sources of $NO_2$ emissions are industrial activities, commercial activities, and transport [35]. Research on the health risks related to air quality has provided information that supports atmospheric health-risk assessments and forms the basis of advice to governments on the measures necessary to protect public health [36]. On the European continent, the air quality index is regularly monitored under the auspices of the EEA.

In 2019, a new Air Quality Index for the Czech Republic was proposed in cooperation with the State Institute of Public Health. This change has led to a more accurate assessment of the current state of air quality and related health impacts. Unlike the previous one, the new index is calculated not from hourly measurements but from the moving average of three-hour concentrations of pollutants. Specifically, these are the concentrations of suspended particulate matter $PM_{10}$, $NO_2$, sulfur dioxide ($SO_2$), and ground-level ozone ($O_3$).

### 2.2. The COVID-19 Pandemic and Transport

The first human coronaviruses (HCoVs) were identified in the 1960s [37]. COVID-19 is the third known coronavirus after SARS-CoV and MERS-CoV [38]. COVID-19 is a novel coronavirus first detected in Wuhan, China, in late 2019 [39]. On 11 March 2020, the World Health Organization (WHO) declared the Coronavirus Disease-2019 (COVID-19) a global pandemic [40]. This pandemic is the most significant global crisis since the Second World War [41].

The COVID-19 pandemic first affected the Czech Republic in spring (from 12 March 2020 to 17 May 2020) and again in the autumn (from 5 October 2020 to 11 April 2021). A state of emergency was declared during these periods. Measures associated with the state of emergency varied according to the situation at the time. Schools were closed at all levels. Restaurants and shops were closed (except for grocery stores, pharmacies, drugstores, and gas stations). Workers who were able to do so due to the character of their work worked from home. The COVID-19 pandemic affected the whole world over time, and severe measures have been taken, including instructing people to isolate from the rest of the population. The measures put in place have significantly reduced everyday mobility, affecting the transport sector [42–47]. Hiselius and Arnfalk note that self-isolation and travel restrictions dramatically reduced the demand for passenger transport, including public transport, as potential passengers were concerned about being infected by other travelers [48]. Various researchers have studied the fall in mobility during lockdowns

due to COVID-19 pandemic around the world [49], including in Spain [50], Sweden [51], Germany [44], France, Italy, China [6–10], and India [45], among others.

The lockdowns resulted in drastic decreases in concentration levels of air pollutants such as $NO_2$, $PM_{2.5}$, $PM_{10}$, CO, GHGs [6–8,10,52]. Muhammad et al. observed a reduction of approximately 20–30% in $NO_2$ emissions in countries such as China, the USA, Spain, France, and Italy during their respective lockdowns [11]. This finding conforms with that of Silver et al., who found that the largest reductions occurred in $NO_2$ emissions, with concentrations 27% lower on average across China [12]. Sharma et al. found that concentrations reduced by 18% for $NO_2$, 10% for CO, 31% for $PM_{10}$, and 43% for $PM_{2.5}$ [53]. As mentioned above, a decline in air pollutants was observed in Europe [54–56], including in the Czech Republic [57–59]; however, it should be noted that there is limited research on the cities of the Czech Republic.

*2.3. Sustainable Transport*

Sustainable development is perhaps the essential idea of our present time [60]. Responding to the challenges of climate change and energy security, people, institutions, and governments at all levels are seeking ways to transition towards more sustainable energy systems using new energy sources and technologies [61]. Reducing car use, traffic congestion, $CO_2$ emissions, and transport-induced stresses on the environment and public health is critical in transitioning toward sustainable urban futures [62]. Transport is responsible for almost 25% of global energy-related greenhouse gas emissions, a share that has been shown to be increasing [29]. Sustainable urban transport is one of the essential elements of sustainable development, as transport-based carbon emissions constitute a significant cause of air quality problems [63]. Several modern cities around the globe are suffering increasing operational complexities as their populations grow and new transport systems are considered, from bicycles to underground systems and car-sharing services [64]. The National Transport Strategy of Scotland proposed the following pyramid for prioritizing sustainable transport (see Figure 1).

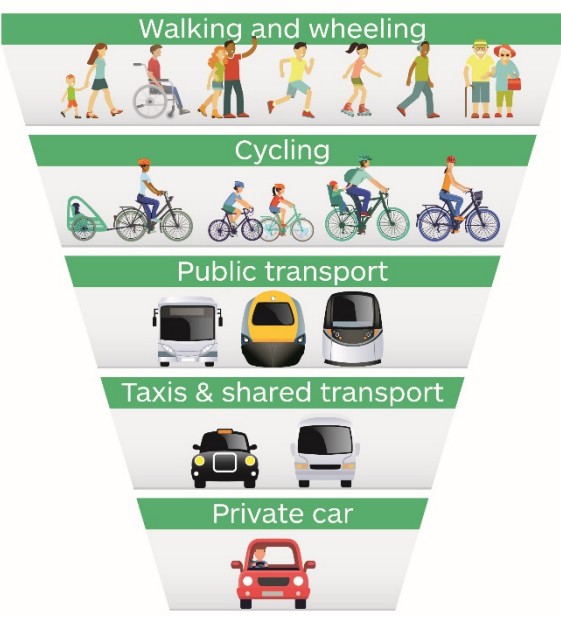

**Figure 1.** Prioritizing sustainable transport [65].

The desire to solve environmental transport problems has risen in various cities. As a result, various Traffic Demand Management policies have been adopted, such as the Low Emissions Zones implemented in Europe [66]. Stockholm and Gothenburg are two Swedish cities that have implemented sustainable transport measures [67]. Another non-European

example is Seoul. Recently, Seoul has been pursuing eco-friendly transport policies to reduce emissions and improve the environment, including restrictions on cars and the introduction of shared bicycles [68].

Mobility and transport policies must achieve more efficient and sustainable transport modes while guaranteeing road safety [42]. Traffic planning should focus on the promotion of public transport to ensure sustainability [69]. There are several ways to move towards sustainable urban transport. Kimbrell notes three key innovations that are seen as promising in terms of achieving a sustainable mobility transition: electric vehicles, shared mobility, and autonomous vehicles [61]. Abduljabbar et al. state that the value of micro-mobility solutions for cities represents a shift towards low-carbon and sustainable modes of transport. They can be a positive force in disrupting private vehicle use, especially for short-distance travel [70]. Electric vehicles (EVs) are a clear example of the change towards using more efficient and environmentally friendly means of transport [71]. Buses also have a vital role in this, as they optimize the use of limited road space by carrying more passengers than personal vehicles [69].

## 3. Current State

This chapter presents the current state of traffic at the selected traffic intersection where pollutant measurements were performed. It is an intersection where there is a lot of air pollution and there is, therefore, a transport station for measuring pollutants. This station measures pollutants such as NOx, $NO_2$, CO, and $PM_{10}$.

A microscopic simulation was created using PTV Vissim software. The simulation represents the current state of operation and also shows the location of the measuring station (see Figure 2).

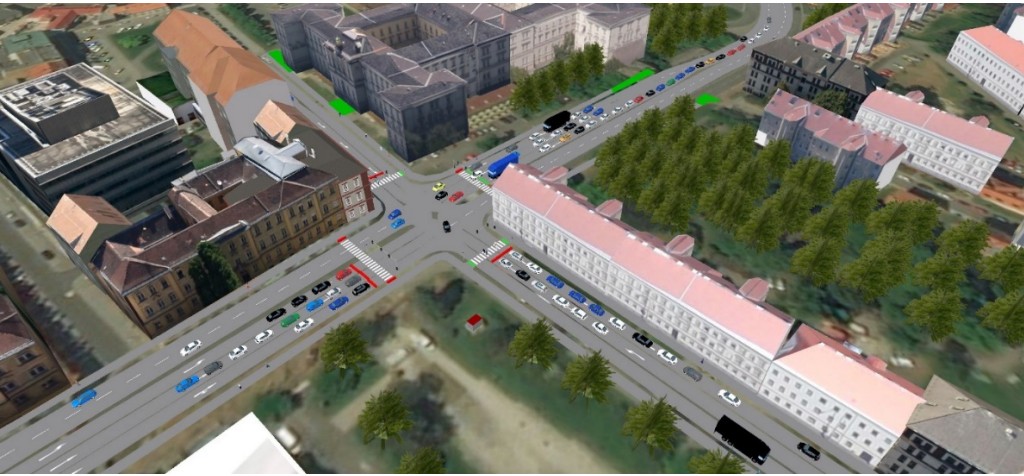

**Figure 2.** Current state of the selected intersection—3D model.

Figure 2 shows the current state of traffic at the selected intersection. This simulation used the annual average of daily traffic intensity during workdays. As can be seen, lines of traffic are forming from the right side. In rush hour, the intersection becomes crowded with vehicles and thus the level of pollutants that are released into the air increases. The volume at the intersection is on average 1100 vehicles per hour; however, the peak-hour traffic volume is 2300 vehicles per hour. More detailed statistics are presented in Table 1.

**Table 1.** Traffic count at the selected intersection [72].

| Traffic Counting | Timing | Intensity |
|---|---|---|
| Annual average of daily traffic intensity—all days | vehicle/day | 23,413 |
| Annual average of daily traffic intensity—workdays | vehicle/day | 25,413 |
| Annual average of daily traffic intensity—free days | vehicle/day | 18,412 |
| Fifty times traffic intensity | vehicle/hour | 2477 |
| Peak-hour traffic intensity | vehicle/hour | 2295 |

Table 1 shows the traffic counts at the selected intersection.

Figure 3 shows a closer view of the intersection, emphasizing the air pollution measuring station (red arrow). As can be seen, the air pollution station is located near the intersection, so the intensity of traffic affects the levels of pollutants that the station captures.

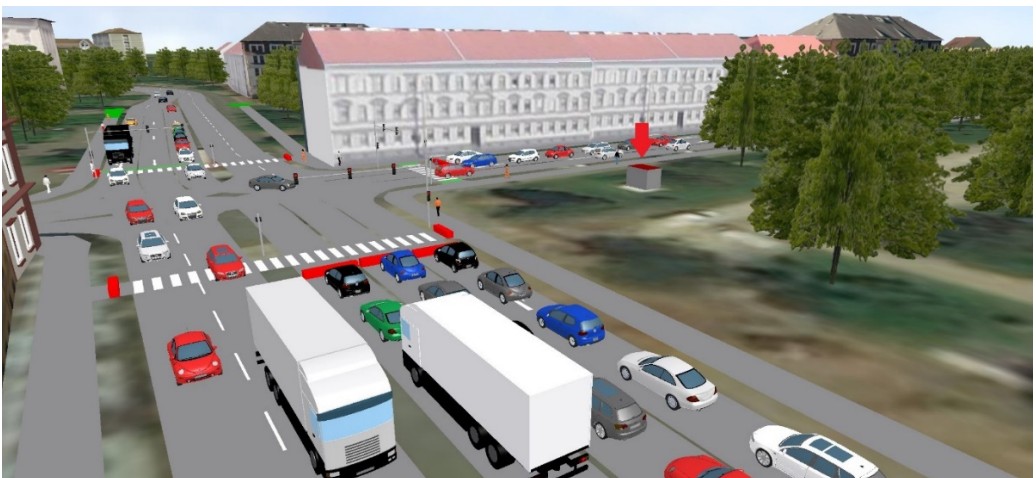

**Figure 3.** The intersection with the air pollution station—3D model.

## 4. Methodology

The data were automatically measured at a monitoring station named ZUHRA in Uherske Hradiste, Czech Republic. This station has been located at the intersection since 2003 and has measured increased air pollution over the years. The measurement is carried out using an automated measuring program, the EUROAIRNET measuring network [73]. Figure 4 shows a photograph of ZUHRA station.

Different measurement methods and instruments are used to measure individual pollutants, such as $NO_x$, CO, and $PM_{10}$. Based on the previous literature review, we can state that the greatest decreases in pollutants were in $NO_2$ and $PM_{10}$. For this purpose, the present study focused on these pollutants for data processing.

$NO_x$ pollutants (including $NO_2$) are measured by chemiluminescence. The principle of the process is based on the excitation of nitrogen molecules by ozone. During the transition of molecules from the excited to the ground energy state, radiation is released in the form of chemiluminescence, which is detected by a photomultiplier. The instrument's design was modified to provide information on the concentrations of nitrogen oxides ($NO_x$) including $NO_2$ [74]. Teledyne API's Model T200 and Model T200U $NO/NO_2/NO_x$ Analyzers use chemiluminescence detection for this measurement (see Principles of Operation, Section 6, in its manual), coupled with state-of-the-art microprocessor technology to provide the sensitivity, stability, and ease of use needed for ambient or dilution CEM monitoring requirements for nitric oxide (NO), $NO_2$, and total $NO_x$. Along with providing high accuracy and dependability, these instruments track operational parameters and issue warnings if they fall outside diagnostic limits, and store easily retrievable data. Proprietary software allows configurable data acquisition capability that can be triggered conditionally or periodically, enabling operators to perform predictive diagnostics and enhanced data

analysis by tracking parameter trends [75]. Interventional studies involving animals or humans, and other studies that require ethical approval, must list the authority that provided approval and the corresponding ethical approval code.

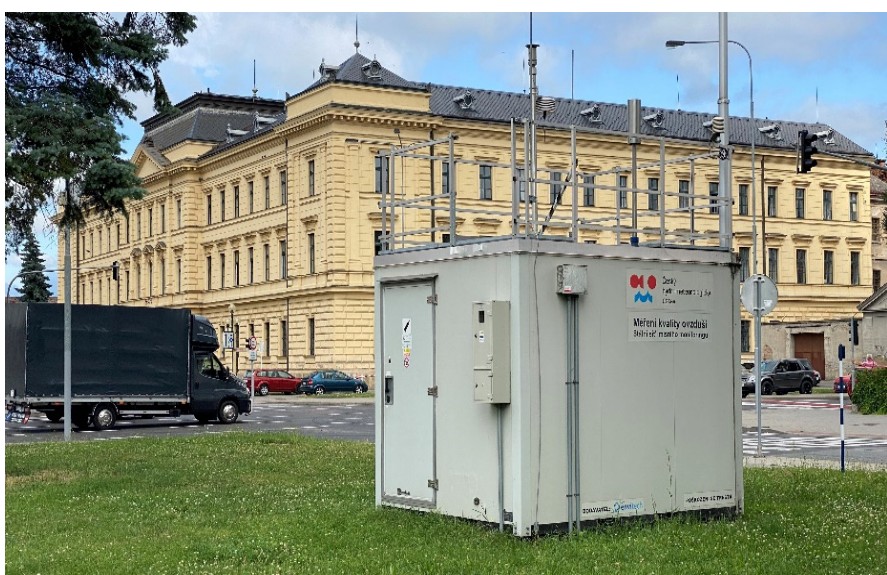

**Figure 4.** ZUHRA traffic station.

$PM_{10}$ pollutants are measured based on the RADIO method (radiometry—beta radiation absorption). This method is based on the absorption of beta radiation in a sample trapped on a filter material. From the difference in beta absorption between exposed and unexposed filter material, which is proportional to the weight of the entrapped dust aerosols, an indication of its concentration is derived [67]. Environment SA, MP101M, is used at the station for this measurement. The MP101M is a continuous monitor that measures suspended particulate matter (SPM) in ambient air. It allows regulatory oversight of $PM_{10}$ and $PM_{2.5}$, with an alarm that is triggered if the threshold is exceeded. The analyzer is based on the measurement of beta-ray attenuation; it determines the fine dust concentration by measuring the amount of radiation that a sample collected on a fiber tape absorbs when exposed to a radioactive source. Low-energy beta rays are absorbed by collision with electrons, whose number is proportional to density. Absorption is thus a function of the mass of the irradiated material, independently of its physicochemical nature. Combined with the optional OPM (Optical Particulate Monitor) module, the monitor allows a precise and real-time evaluation of $PM_{10}$, $PM_{2.5}$, and $PM_1$ simultaneously. The CPM module can be plugged additionally into all existing MP101Ms between the RST sampling line and the MP101M monitor [76].

The data analyzed in this study on $PM_{10}$ and $NO_2$ concentrations were generated through hourly measurements, collected from December 2019 to December 2020. The data were provided by the Czech Hydrometeogrological Institute (CHMI) [77]. To create more homogeneous datasets for statistical analysis, the measurements were grouped into smaller periods—monthly intervals. Depending on the number of days in the relevant month, the intervals contained 672, 720, and 744 measurements. Statistical analysis was performed on the obtained data; the results were graphically visualized and evaluated by means of exploratory data analysis. Monthly descriptive statistics on $PM_{10}$ and $NO_2$ concentrations were calculated from all measured data, excluding months where valid data were not provided (i.e., April and May 2019, during which months there were missing values for concentrations of $PM_{10}$). Also, results from the automatic calibration of the measuring devices were removed from the datasets. Before the statistical analysis, normality tests were performed using skewness and kurtosis coefficients [78]. The data normality was also verified using Q-Q plots and histograms.

Although the distribution of measured $PM_{10}$ and $NO_2$ concentrations in some monthly intervals was not normal but symmetrical (which was verified by means of Q-Q plots and histograms) and included outliers, further analysis was carried out assuming a normal distribution. The datasets were large in scope, so it was possible to do this based on asymptotic normality. To compare the level of concentrations of these pollutants in individual months when states of emergency were declared, a parametric approach was utilized (two-sample *t*-test) [79]. Only temporal intervals with complete data pairs for both months were used for the statistical evaluation.

## 5. Results

First, a basic analysis of the datasets was performed. Invalid measurements and values that the automatic evaluation software marked as incorrect were removed. An exploratory data analysis of the whole measurement campaign was conducted as a second step. $PM_{10}$ and $NO_2$ concentrations in individual years were graphically visualized—see Figures 5 and 6, where blue denotes concentrations in 2019 and red denotes concentrations in 2020. Figures 5 and 6 make it visually apparent that $NO_2$ and $PM_{10}$ concentrations were lower in 2020 than in 2019. Figure 6 shows that the measured concentrations of $NO_2$ were systematically lower throughout 2020 compared to 2019. However, for $PM_{10}$, this was not the case: $PM_{10}$ concentrations were not systematically lower throughout 2020. The most significant decreases of measured $PM_{10}$ concentrations in 2020 compared to 2019 are evident in February, April, May, June, October, and December—see Figure 5. The assessment of the deviation in measured concentrations in individual months is the subject of the following analysis.

Another graphical comparison of measurement pairs in individual months was performed using box-and-whisker plots. Figures 7 and 8 show all monthly comparisons of $PM_{10}$ and $NO_2$ concentrations in 2019 (left box-and-whisker plot) and 2020 (right box-and-whisker plot). For the sake of greater clarity, the individual months were separated from each other. The time periods marked in green represent the periods when a state of emergency was declared in the Czech Republic.

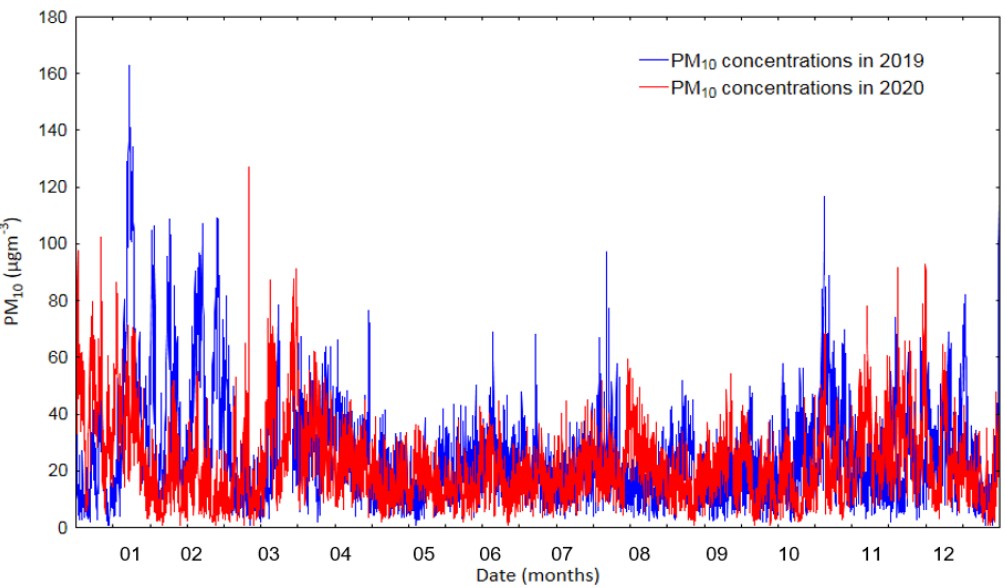

**Figure 5.** Graphical visualization of $PM_{10}$ concentration levels in 2019 (blue) and 2020 (red).

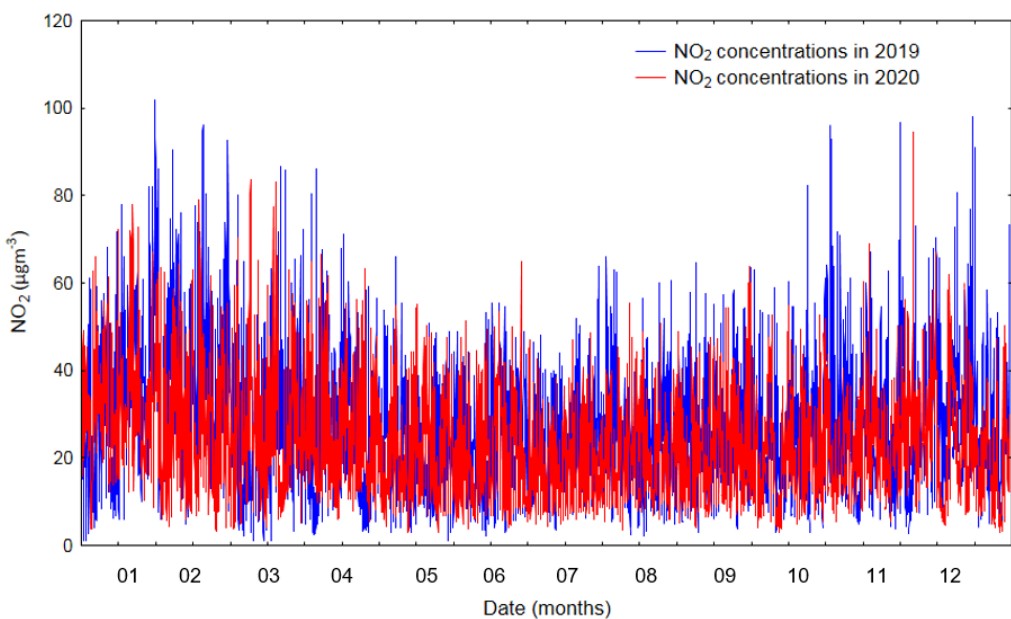

**Figure 6.** Graphical visualization of $NO_2$ concentration levels in 2019 (blue) and 2020 (red).

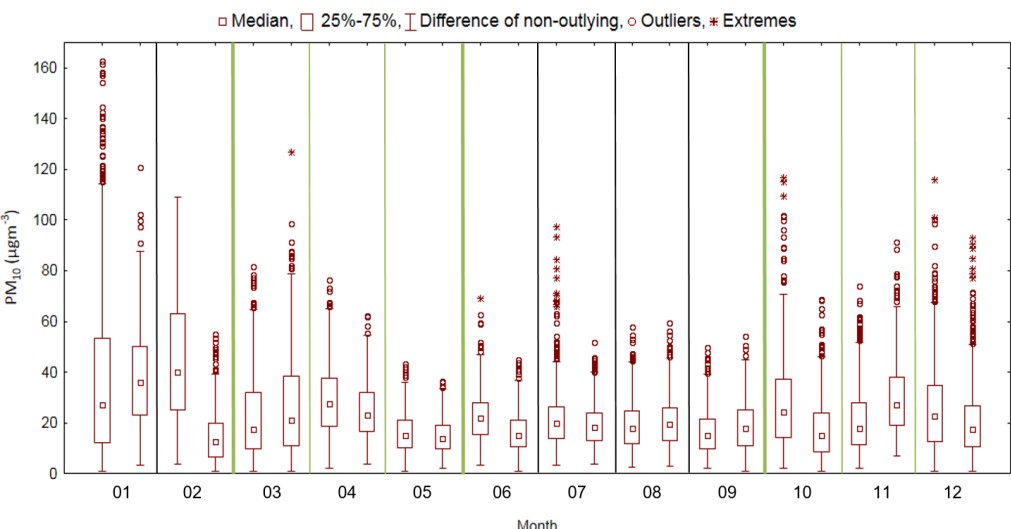

**Figure 7.** Box-and-whisker-plots for monthly $PM_{10}$ concentrations in 2019 (**left**) and 2020 (**right**).

Figure 7 shows box-and-whisker plots for monthly $PM_{10}$ concentrations in 2019 and 2020. As can be seen, the distribution of monthly $PM_{10}$ measurements did not show any significant asymmetry around the median, except for in January and February 2019, and the month of March in both years. The median $PM_{10}$ values changed over time. For example, it can be observed that during January, there was a significant increase in $PM_{10}$ concentrations in 2020 compared to 2019. At the same time, this comparison clearly shows there was lower measurement variability in 2020. On the contrary, in February, there were significantly lower $PM_{10}$ concentrations in 2020 compared to the same month in 2019 and greater measurement variability was evident in 2019. By comparing the medians of the $PM_{10}$ measurements, it can be seen that that the measured concentrations were similar in both years during July and August.

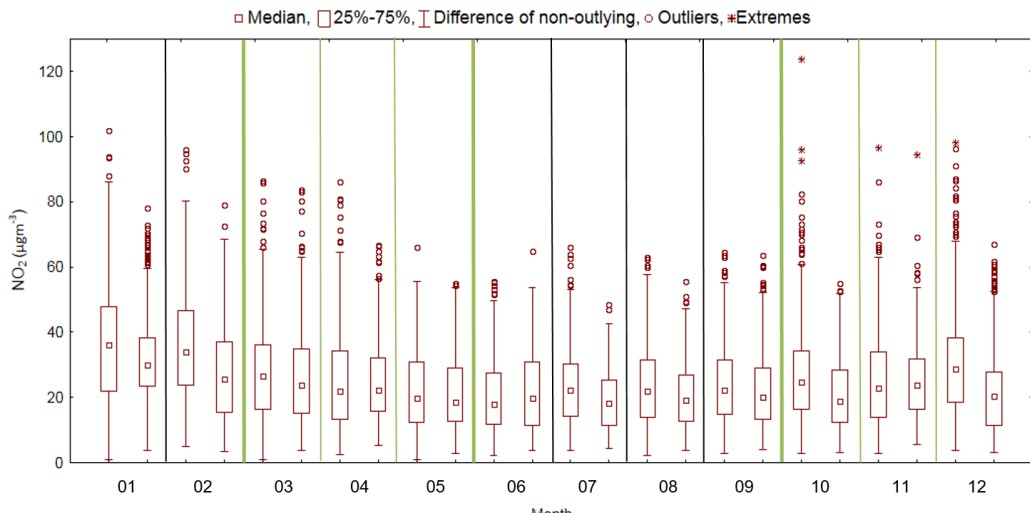

**Figure 8.** Box-and-whisker-plots for monthly $NO_2$ concentrations in 2019 (**left**) and 2020 (**right**).

Figure 8 shows box-whisker-plots for monthly $NO_2$ concentrations in 2019 and 2020. As can be seen, the median $NO_2$ values were higher in all months in 2019 than in 2020, except for June and November. In April, the concentrations were practically identical in both years. The median $NO_2$ concentration changed over time. It can be observed that during January, February, October, and December, there was a significant decrease in $NO_2$ concentrations in 2020 compared to 2019; at the same time, this comparison clearly shows lower measurement variability in 2020.

The descriptive statistics on monthly $PM_{10}$ concentrations, specifically the mean, median, minimum, and maximum value, and standard deviation, are listed in Table 2. The average $PM_{10}$ concentration was 25.43 µg m$^{-3}$ in 2019 and 21.82 µg m$^{-3}$ in 2020. The lowest monthly mean (14.70 µg m$^{-3}$) for $PM_{10}$ was measured in May 2020, while the highest monthly mean (44.80 µg m$^{-3}$) was measured in February 2019. The highest value (162.90 µg m$^{-3}$) was measured in January 2019, and the lowest values (1.00 µg m$^{-3}$) were measured in January, March, May, and December 2019, and in February, March, June, September, October, and December 2020 (see Table 2).

**Table 2.** Descriptive statistics of the measured $PM_{10}$ concentrations in µg m$^{-3}$.

| $PM_{10}$ | Mean | | Median | | Min | | Max | | SD | |
|---|---|---|---|---|---|---|---|---|---|---|
| Month | 2019 | 2020 | 2019 | 2020 | 2019 | 2020 | 2019 | 2020 | 2019 | 2020 |
| January | 34.74 | 37.82 | 27.20 | 36.05 | 1.00 | 3.20 | 162.90 | 102.82 | 35.06 | 20.36 |
| February | 44.80 | 14.81 | 40.00 | 12.60 | 3.80 | 1.00 | 109.00 | 55.10 | 24.62 | 10.35 |
| March | 22.50 | 26.93 | 17.20 | 20.90 | 1.00 | 1.00 | 81.60 | 126.90 | 17.06 | 20.38 |
| April | 28.60 | 24.96 | 27.60 | 23.05 | 2.00 | 3.60 | 76.40 | 62.30 | 13.60 | 10.92 |
| May | 16.12 | 14.70 | 14.90 | 16.65 | 1.00 | 2.10 | 43.40 | 36.20 | 8.12 | 6.98 |
| June | 22.36 | 16.33 | 21.80 | 15.00 | 3.40 | 1.00 | 69.00 | 44.70 | 9.53 | 8.07 |
| July | 21.57 | 17.78 | 19.70 | 18.10 | 3.40 | 3.80 | 97.20 | 51.90 | 12.04 | 7.90 |
| August | 19.07 | 20.56 | 17.70 | 19.30 | 2.40 | 2.70 | 57.90 | 59.20 | 9.64 | 10.07 |
| September | 16.31 | 18.24 | 14.90 | 17.70 | 2.20 | 1.00 | 49.80 | 54.30 | 8.43 | 9.61 |
| October | 27.87 | 17.45 | 24.25 | 14.80 | 2.00 | 1.00 | 116.60 | 68.60 | 18.35 | 11.40 |
| November | 21.25 | 29.40 | 17.80 | 27.20 | 2.10 | 7.00 | 73.90 | 91.40 | 12.92 | 14.00 |
| December | 25.98 | 21.24 | 22.80 | 17.40 | 1.00 | 1.00 | 115.80 | 92.90 | 17.49 | 15.22 |

Table 3 shows the descriptive statistics for the monthly $NO_2$ concentrations. The average $NO_2$ concentration was 26.65 µg m$^{-3}$ in 2019 and 23.49 µg m$^{-3}$ in 2020. The lowest monthly mean (19.06 µg m$^{-3}$) of $NO_2$ was measured in July 2020, while the highest monthly mean (35.98 µg m$^{-3}$) was measured in February 2019. The highest value (123.80 µg m$^{-3}$)

was measured in October 2019, and the lowest values (1.00 µg m$^{-3}$) were measured in January, March, and May 2019 (see Table 3).

**Table 3.** Descriptive statistics on the measured NO$_2$ concentrations in µg m$^{-3}$.

| NO$_2$ | Mean | | Median | | Min | | Max | | SD | |
|---|---|---|---|---|---|---|---|---|---|---|
| Month | 2019 | 2020 | 2019 | 2020 | 2019 | 2020 | 2019 | 2020 | 2019 | 2020 |
| January | 35.94 | 31.62 | 36.20 | 30.00 | 1.00 | 3.80 | 101.80 | 78.00 | 17.73 | 12.38 |
| February | 35.98 | 27.44 | 34.00 | 25.60 | 5.00 | 3.30 | 96.20 | 79.00 | 16.72 | 14.28 |
| March | 27.33 | 26.06 | 26.60 | 23.70 | 1.00 | 3.80 | 86.50 | 83.60 | 14.75 | 14.02 |
| April | 25.18 | 25.01 | 22.00 | 22.20 | 2.50 | 5.40 | 86.10 | 66.60 | 15.17 | 12.00 |
| May | 22.22 | 21.35 | 19.90 | 18.60 | 1.00 | 2.90 | 66.00 | 55.10 | 12.02 | 10.94 |
| June | 20.79 | 21.64 | 18.00 | 19.70 | 2.10 | 3.80 | 55.50 | 64.80 | 11.37 | 11.65 |
| July | 23.43 | 19.06 | 22.30 | 18.30 | 3.60 | 4.40 | 66.00 | 48.60 | 10.99 | 8.83 |
| August | 23.11 | 20.64 | 21.80 | 19.30 | 2.30 | 3.60 | 63.10 | 55.50 | 11.65 | 9.90 |
| September | 24.35 | 22.03 | 22.20 | 20.10 | 2.90 | 4.00 | 64.70 | 63.70 | 11.70 | 11.24 |
| October | 26.68 | 20.82 | 24.70 | 18.90 | 2.90 | 3.10 | 123.80 | 54.90 | 14.56 | 10.63 |
| November | 25.24 | 24.72 | 22.80 | 23.70 | 2.70 | 5.50 | 96.60 | 94.50 | 14.55 | 10.49 |
| December | 30.09 | 21.74 | 27.70 | 20.30 | 3.60 | 3.10 | 98.10 | 67.00 | 15.88 | 12.61 |

To compare the levels of PM$_{10}$ and NO$_2$ in individual months when states of emergency were declared, a parametric two-sample *t*-test was used. The test was performed at the significance level of $\alpha$ = 0.05. The results are shown in Table 4. In the first column of this table, the individual pollutants are listed. In the second column, the alternative test hypothesis of variance match, AH$_F$, is shown. The test statistics of variance match test, F, is given in next column, followed by its corresponding *p*-value. In the fifth column, the alternative hypothesis of mean values match, AH$_t$, is presented, and the column denoted *t* contains the results of the mean values match test. Finally, the last column contains the corresponding *p*-value for the two-sample *t*-test.

**Table 4.** Results of the two-sample *t*-test for PM$_{10}$ a NO$_2$ concentrations in 2019 and 2020 when states of emergency were declared.

| | AH$_F$ | F | *p*-Value | AHt | *t* | *p*-Value |
|---|---|---|---|---|---|---|
| PM$_{10}$ | $\sigma_1^2 \neq \sigma_2^2$ | 1.072 * | 0.005 * | $\mu_1 > \mu_2$ | 4.228 * | 0.000 * |
| NO$_2$ | $\sigma_1^2 \neq \sigma_2^2$ | 1.074 * | 0.005 * | $\mu_1 > \mu_2$ | 5.414 * | 0.000 * |

* indication of the values of relevant statistics, when the related null hypothesis was rejected at the 5% significance level.

It follows from the results of the *t*-test that measured concentrations of PM$_{10}$ and NO$_2$ were higher in 2019 than in 2020. It is clear from the analysis that the COVID-19 pandemic and the associated declared states of emergency had a significant impact on reducing traffic, which led to reduced air pollution from these pollutants at the intersection in Uherske Hradiste.

## 6. Discussion

This paper presents the results from research evaluating the impacts of COVID-19 pandemic on air pollution—specifically on the pollutants PM$_{10}$ and NO$_2$—at a traffic intersection in Uherske Hradiste, in the Czech Republic. For this purpose, data from CHMI was used. The performed analysis revealed statistically significant differences between the levels of PM$_{10}$ and NO$_2$ during periods in which a states of emergency had been declared. The results of the two-sample *t*-test indicate, at a significance level of $\alpha$ = 0.05, that for both pollutants, the measured concentrations were lower in individual months when states of emergency were declared in 2020 than in 2019. On average, NO$_2$ and PM$_{10}$ concentrations decreased by 2.70 µg m$^{-3}$ and 2.05 µg m$^{-3}$, that is, by 11.00% and 9.23%, respectively. The decrease in pollutant concentrations was lower that that found in other countries in the same period. States such as China, the USA, Spain, France, and Italy NO$_2$ production

reduced by 20–30% during the COVID-19 pandemic [11,12,53]. A similar observation can be made when comparing findings on $PM_{10}$ concentrations: again, we found a smaller decline than that recorded in other countries.

There was also a significant variability in $PM_{10}$ concentrations throughout the year, with much higher values noticable in the winter months. This is most likely due to the variability in $PM_{10}$ particle sources throughout the year. In cold conditions, the intensity of heating increases and becomes a significant source of air pollution. A significant source of suspended particles is traffic at this location. It is a stable source in that it is relevant both in summer and in winter, because vehicles produce $PM_{10}$ particles by resuspension and due to the abrasion of brake pads, clutches, tires, and the road surface. Furthermore, it is clear that the variability observed in $NO_2$ concentrations was not as the variability in $PM_{10}$ particles through the year. However, the variability in $NO_2$ levels was also highest in the winter months, that is, in January, February, March, November, and December in 2019.

It should be noted that this research was been primarily on a selected intersection in the city of Uherske Hradiste, Czech Republic. However, the findings do not necessarily apply to all cities in the Czech Republic. Further research will be focused on the other pollutants that the station measures. There will also be evaluations conducted at different stations in the Czech Republic and a mutual comparison between them. Inhabitants should learn from the COVID-19 pandemic and reduce urban traffic. In general, cities should discourage people from using cars by creating sustainable transport options.

## 7. Conclusions

The paper verified the hypothesis that concentrations of $PM_{10}$ and $NO_2$ pollutants would be lower during individual months when states of emergency related to the COVID-19 pandemic were declared. CHMI provided the necessary data for 2019 and 2020. For the research, two pollutants, $PM_{10}$ and $NO_2$, were selected. These pollutants were chosen based on a literature review of research in other countries, which found that these pollutants were the ones that fell the most during the COVID-19 pandemic. Based on our results, we conclude that the reduction of transport impacted air pollution during the COVID-19 pandemic in Uherske Hradiste, Czech Republic, by reducing it.

A combination of basic statistical methods was used to compare $PM_{10}$ and $NO_2$ concentrations and the temporal variability was evaluated. The statistical evaluation led to significant findings regarding the decrease in $PM_{10}$ and $NO_2$ pollutants concentrations during the COVID-19 pandemic. The statistical approach also served as an additional control for data quality, because there were errors in data transmission for $PM_{10}$ concentrations from the analyzers in the diagnostic system (in April and May 2019). Statistical data analysis was performed using STATISTICA software.

The concentration of $NO_2$ was consistently lower in 2020 compared to 2019 (for every month except June). This was not the case for $PM_{10}$; in this case there was variability over several months. The population must learn from the situation and start using sustainable urban transport systems. For example, using shared cars, urban public transport, shared bicycles are all ways to reduce concentrations of $PM_{10}$ and $NO_2$ in urban air.

**Author Contributions:** Conceptualization, K.V.; methodology, K.V. and P.V.; software, R.D. and P.V.; validation, K.V. and P.V.; formal analysis, K.V.; investigation, P.V., K.V. and R.H.; resources, K.V., P.V. and R.H.; data curation, P.V.; writing—original draft preparation, K.V., P.V., R.H. and R.D.; writing—review and editing, K.V., P.V. and R.H.; visualization, K.V., P.V. and R.H.; project administration, K.V.; funding acquisition, K.V. All authors have read and agreed to the published version of the manuscript.

**Funding:** Thisresearch was funded by grant RVO/FLKŘ/2021/03 and IGA/FLKŘ/2021/01 of Tomas Bata University in Zlín.

**Data Availability Statement:** The data presented in this study are available on request from the corresponding author.

**Acknowledgments:** This research was supported by the project RVO/FLKŘ/2021/03 and IGA/FLKŘ/ 2021/001 of Tomas Bata University in Zlín.

**Conflicts of Interest:** The authors declare no conflict of interest.

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
