# Peer review of "Road Transport and Its Impact on Air Pollution during the COVID-19 Pandemic"

_sustainability, doi:10.3390/su132111803_

Round 1
Reviewer 1 Report
The paper analyses monthly data about concentration of NO2 and PM10 in 2019 and 2020. Graphs and statistical t-tests were adopted to highlight reductions of these pollutants in 2020. The topic is within the scope of the journal, however the analysis techniques are very simple and several assumptions need to be clarified before publication.
Chapter 1
Page 1: the authors should add a paragraph describing the evolution of pandemic in 2020 in Czech Republic, as well as countermeasures adopted by local government to contain the spread of the virus (if any). In this way the reader will be able to understand potential changes on reported monthly concentrations of pollutants.
Chapter 3
Page 3: why a traffic microsimulation software was used if traffic data were not reported in the paper?
Chapter 4
Page 7, line 251-252: the authors should clarify how they aggregated hourly measures into monthly data.
Page 7, line 255: the authors should clarify what they mean with “valid data”. In addition, why did they retain months with not valid data?
Page 7, line 263: I suggest the authors to explain why a parametric tests was adopted even if data are not normally distributed
Chapter 5
Page 8: I suggest the authors to replace the label of the horizontal axes with “Date (months)”
Page 9: I cannot understand which is the left and right side of box-plots in figure 7 and 8.
Page 11, line 333-334: the authors often referred to a period “when states of emergency were declared”, however they have never indicated the specific range.
Chapter 7
Page 12, line 392: The authors wrote that they “took only transport into account, not other sources of PM10 pollution”, however there are no references to traffic data in the paper. I suggest to clarify this sentence.
Author Response
Attached in docx.

Reviewer 2 Report
The topic of the assessed article is current, it is appropriate to include it in the special issue of the journal. The comparative analysis presented in the article is a good basis, for publishing, some parts will need to be improved. The authors focused on a narrow area of air pollution, I recommend changing the title of the article to "Road transport ...". In the introduction to the article, explain in what research the article was prepared and define the gaps that need to be addressed. Beware of the incorrect surname not Hotorova but Hoterova. Theoretical background is standard, I recommend adding other sources about the decrease in air pollution in the Czech Republic.
Part of the methodology significantly narrowed the whole solution to NO2 and PM10 at one specific intersection. I recommend better explaining how significantly road traffic affects the state of NO2 and PM10 and what effect heating has in the vicinity of the intersection. In the results section, I recommend improving the explanation for each of the figures 5 -8.
In the discussion, I recommend adding a broader context to the topic. I recommend adding other relevant ideas and recommendations to the conclusion. Be careful when formatting references.
Author Response
Attached in docx.
Round 2
Reviewer 1 Report
The authors addressed all the comments and they significantly improved the quality of the paper.
Author Response
Thank you
Reviewer 2 Report
The second version of the article was improved by the authors. I appreciate the narrowing of the title of paper to road transport, the addition of the project within which the research was solved, the inclusion of a paragraph explaining the specific course of the state of emergency.
From a professional point of view, it is important to include the average and peak permeabilities of a given intersection, plus the inclusion of new references. The graphs included in the article were explained in detail in a suitable way. In the current version of the article, however, it is necessary to correct the formatting of references.
Author Response
All of the points from review report was added to the paper.